# Disulfide stabilization reveals conserved dynamic features between SARS-CoV-1 and SARS-CoV-2 spikes

Xixi Zhang[1,*], Zimu Li[1,*], Yanjun Zhang[2,*], Yutong Liu[1,*], Jingjing Wang[1], Banghui Liu[1], Qiuluan Chen[3], Qian Wang[2], Lutang Fu[4], Peiyi Wang[4], Xiaolin Zhong[3], Liang Jin[3], Qihong Yan[1,2], Ling Chen[1,2,5], Jun He[1], Jincun Zhao[2,5], Xiaoli Xiong[1]

SARS-CoV-2 spike protein (S) is structurally dynamic and has been observed by cryo-EM to adopt a variety of prefusion conformations that can be categorized as locked, closed, and open. S-trimers adopting locked conformations are tightly packed featuring structural elements incompatible with RBD in the "up" position. For SARS-CoV-2 S, it has been shown that the locked conformations are transient under neutral pH. Probably because of their transience, locked conformations remain largely uncharacterized for SARS-CoV-1 S. In this study, we introduced x1, x2, and x3 disulfides into SARS-CoV-1 S. Some of these disulfides have been shown to preserve rare locked conformations when introduced to SARS-CoV-2 S. Introduction of these disulfides allowed us to image a variety of locked and other rare conformations for SARS-CoV-1 S by cryo-EM. We identified bound cofactors and structural features that are associated with SARS-CoV-1 S locked conformations. We compare newly determined structures with other available spike structures of SARS-related CoVs to identify conserved features and discuss their possible functions.

## Introduction

Betacoronavirus subgenus *Sarbecovirus* is composed of SARS-CoV-1, SARS-CoV-2, and other severe acute respiratory syndrome-related coronaviruses (SARSr-CoVs) (Coronaviridae Study Group of the International Committee on Taxonomy of Viruses, 2020). Because of SARS-CoV-1's known ability to infect humans and the SARS-CoV-2 pandemic, Sarbecoviruses are subjected to intense research. Among structural proteins of Sarbecoviruses, spike (S) protein is responsible for receptor binding and mediates membrane fusion between cell and virus to initiate infection (Li, 2016). Being the most exposed protein on the virus surface, it is the main target of the immune system. Most available SARS-CoV-2 vaccines used S protein or its derivatives as the immunogen (Krammer, 2020) and a number of antibodies targeting the SARS-CoV-2 S protein have been approved for emergency use authorization (Corti et al, 2021). For these reasons, S protein is a major focus for coronavirus vaccine and therapeutics development.

Murine hepatitis virus (Walls et al, 2016a), human coronavirus HKU1 (HCoV-HKU1) (Kirchdoerfer et al, 2016), and human coronavirus NL63 (HCoV-NL63) (Walls et al, 2016b) spikes were the first structurally characterized coronavirus (CoV) spikes. They are all homotrimeric and adopt single conformations with all their three receptor-binding domains (RBDs) in "down" positions. Among Sarbecoviruses, the structures of SARS-CoV-1 S protein were first determined and conformational dynamics was first observed for a CoV S-trimer. RBDs in SARS-CoV-1 spikes were observed to be in the "up" and "down" positions making the spikes exhibit open and closed conformations, respectively (Gui et al, 2017; Yuan et al, 2017). In line with SARS-CoV-1 S, RBD "up" open and RBD "down" closed spikes were observed when SARS-CoV-2 S structures were initially determined (Wrapp et al, 2020b; Walls et al, 2020). However, a third prefusion conformation, designated "locked," was subsequently identified in multiple studies using either full-length spikes or spike ectodomains (Bangaru et al, 2020; Cai et al, 2020; Toelzer et al, 2020; Wrobel et al, 2020; Xiong et al, 2020). S-trimer in locked conformation features three "down" RBDs. Different from the S-trimer in closed conformation which also have three "down" RBDs, locked S-trimer is structurally more ordered, adopting a more tightly packed trimeric quaternary structure, and usually possessing features including, bound lipid in RBD, rigidified Domain D region, and ordered fusion peptide proximal region (FPPR, residues 833–855 in SARS-CoV-2 S) (Xiong et al, 2020; Qu et al, 2022). Interestingly, locked conformation appears

---

[1]State Key Laboratory of Respiratory Disease, CAS Key Laboratory of Regenerative Biology, Guangdong Provincial Key Laboratory of Stem Cell and Regenerative Medicine, Guangdong Provincial Key Laboratory of Biocomputing, Guangzhou Institutes of Biomedicine and Health, Chinese Academy of Sciences, Guangzhou, China [2]State Key Laboratory of Respiratory Disease, Guangzhou Institute of Respiratory Health, First Affiliated Hospital of Guangzhou Medical University, Guangzhou, China [3]Bioland Laboratory (Guangzhou Regenerative Medicine and Health - Guangdong Laboratory), Guangzhou, China [4]Cryo-electron Microscopy Center, Southern University of Science and Technology, Shenzhen, China [5]Guangzhou Laboratory, Guangzhou International Bio Island, Guangzhou, China

Correspondence: xiong_xiaoli@gibh.ac.cn; zhaojincun@gird.cn; he_jun@gibh.ac.cn
Xixi Zhang's present address is Liangzhu Laboratory, Zhejiang University, Hangzhou, China
Qiuluan Chen's present address is Yunzhou Biosciences (Guangzhou) Co., Ltd., Guangzhou, China
*Xixi Zhang, Zimu Li, Yanjun Zhang, and Yutong Liu contributed equally to this work

to be transient, in pH-neutral PBS, only a small fraction of purified S-trimer-adopted locked conformations (Xiong et al, 2020).

Detailed structural analysis of locked SARS-CoV-2 S-trimers identified that interactions between Domain D and Domain C–D hinge region likely restrain RBD movement leading to more compact spike packing (Qu et al, 2022). A comparison with other CoV S structures revealed that HCoV-NL63 (Walls et al, 2016b), murine hepatitis virus (Walls et al, 2016a), infectious bronchitis virus (Shang et al, 2018), and porcine delta coronavirus (Xiong et al, 2018) spikes are structurally more related to locked structures of SARS-CoV-2 spike, they all have ordered FPPR which is incompatible with RBD in the "up" position (Xiong et al, 2020). More perplexingly, locked conformations were not observed for spikes on fixed SARS-CoV-2 virus particles (Ke et al, 2020; Turoňová et al, 2020; Yao et al, 2020).

We and others engineered disulfide bonds to trap SARS-CoV-2 spike in RBD "down" conformations (Henderson et al, 2020; McCallum et al, 2020; Xiong et al, 2020). In addition to increased spike stability, we found that x1 (Xiong et al, 2020) and x3 (Qu et al, 2022) disulfide bonds increase the proportions of purified spikes adopting locked conformations. Structural studies of x3 disulfide-stabilized locked spikes allowed us to further classify locked spikes into the originally identified "locked-1" (Bangaru et al, 2020; Cai et al, 2020; Toelzer et al, 2020; Wrobel et al, 2020; Xiong et al, 2020) and additionally identified "locked-2" (Qu et al, 2022) conformations. The two conformations differ primarily in ways how Domain D region rigidifies. We observed that in the "locked-1" conformation Domain D is rigidified with a large, disordered Domain D-loop, whereas in the "locked-2" conformation Domain D is rigidified with the Domain D-loop fully ordered. Refolding of the Domain D-loop is observed between the two locked conformations resulting in some Domain D residues engaging in different interactions (Qu et al, 2022). Protomers in a SARS-CoV-2 S trimer can adopt exclusively "locked-1" or "locked-2" conformations or a trimer can be formed by protomers of the two locked conformations in various combinations (Qu et al, 2022). Together with "closed" and "open" conformations, prefusion SARS-CoV-2 spike exhibits complex dynamics.

Despite considerable sequence homology to SARS-CoV-2 S, currently determined structures of SARS-CoV-1 spikes are either in closed or open conformation (Gui et al, 2017; Yuan et al, 2017). Locked conformations remain elusive for the SARS-CoV-1 spike, therefore dynamics of the spike likely remains incompletely characterized. In this study, we engineered SARS-CoV-1 spikes bearing the x1, x2, and x3 disulfides. We purified engineered SARS-CoV-1 spikes and characterized their biochemical properties. By Cryo-EM, engineered SARS-CoV-1 spikes allowed us to determine a series of spike structures adopting previously undetected conformations. We compare these structures with available spike structures of other Sarbecoviruses to identify conserved features.

## Results

### Production of SARS-CoV-1 spikes with engineered disulfide bonds

x1, x2, and x3 (sites in SARS-CoV-2 S numbering: x1, S383C, and D985C; x2, G413C, and V987C; x3, D427C, and V987C) disulfide bonds

were designed for the SARS-CoV-2 spike (Xiong et al, 2020; Qu et al, 2022). These disulfides facilitate the formation of covalently linked spike trimers through the introduced disulfides between RBD and S2 of a neighboring spike protomer. x1 disulfide has been previously introduced in the SARS-CoV-1 spike to show successful disulfide formation (McCallum et al, 2020). We introduced these three disulfide bonds in equivalent positions in the SARS-CoV-1 S sequence generating three spike variants: S/x1 (S370C and D967C), S/x2 (G400C and V969C), and S/x3 (D414C and V969C) (Figs 1A and S1). These spike variants were expressed in Expi293 cells to be secreted into culture media. Compared with purified unmodified SARS-CoV-1 spike ectodomain, purified S/x1, S/x2, and S/x3 SARS-CoV-1 spikes successfully form covalently linked trimers in SDS–PAGE under nonreducing conditions and these trimers could be reduced to monomers under denaturing conditions (Figs 1B and S2). x1, x2, and x3 disulfides in the SARS-CoV-1 spike exhibit different sensitivities to the reducing agent. Under native conditions, x2 disulfide is more resistant to reduction by DTT than x1 and x3 disulfides, likely reflecting different local chemical environments for different disulfides (Fig S2). Resistance to DTT reduction was also observed for x2 disulfide in the SARS-CoV-2 spike (Qu et al, 2022). Negative-staining EM of purified spike variants showed that S/x1, S/x2, and S/x3 spikes form trimers with morphologies indistinguishable to an unmodified SARS-CoV-1 spike ectodomain (S/native) (Fig 1C). These results confirmed that x1, x2, and x3 disulfides previously designed for the SARS-CoV-2 spike are compatible with the production of well-formed covalently linked SARS-CoV-1 S-trimers.

### Engineered S/x3 spike induces mouse immune sera

We have previously shown that "x2" disulfide stabilized, RBD "down," SARS-CoV-2 S-R/x2 (S-R meaning that the multibasic S1/S2 furin cleavage site is modified to a single arginine) S-trimer-induced sera exhibit poor correlation with sera raised by S-trimers with unrestrained RBDs in antigen binding, indicating different antibodies are induced among the RBD "down" S-trimer and S-trimers with unrestrained RBDs, although no difference of statistical significance in neutralization potency among the sera was found (Carnell et al, 2021). It remains poorly understood how an RBD "down" S-trimer induces a different antibody response while generating sera of similar neutralization potencies comparing with S-trimers with unrestrained RBDs. As we have previously demonstrated that "x2"-stabilized SARS-CoV-2 S-trimer was able to induce immune sera (Carnell et al, 2021) and "x1"-stabilized S-trimer is heterogenous in spike conformation (Fig 2A–E), purified S/x3 spike was chosen to immunize mice employing a prime-boost immunization strategy. S/x3 was found to exclusively adopt an RBD "down"-locked-1 conformation by cryo-EM (Fig 2H). After priming by 10 μg of S/x3 spike, mouse sera showed induction of neutralization titers against SARS-CoV-1 pseudovirus (Fig 1D). A boost by another dose of 10 μg of S/x3 spike increased neutralization titer further by ~100-fold (Fig 1D). The complete immunization regimen induced a serum neutralization titer of ~5 $\log_{10}$ dilution units (Fig 1D). We found mouse immune sera raised with engineered SARS-CoV-2 RBD "down" S-R/x2 spike protein as the immunogen gave similar neutralization titers (Carnell et al, 2021). By ELISA (enzyme-linked

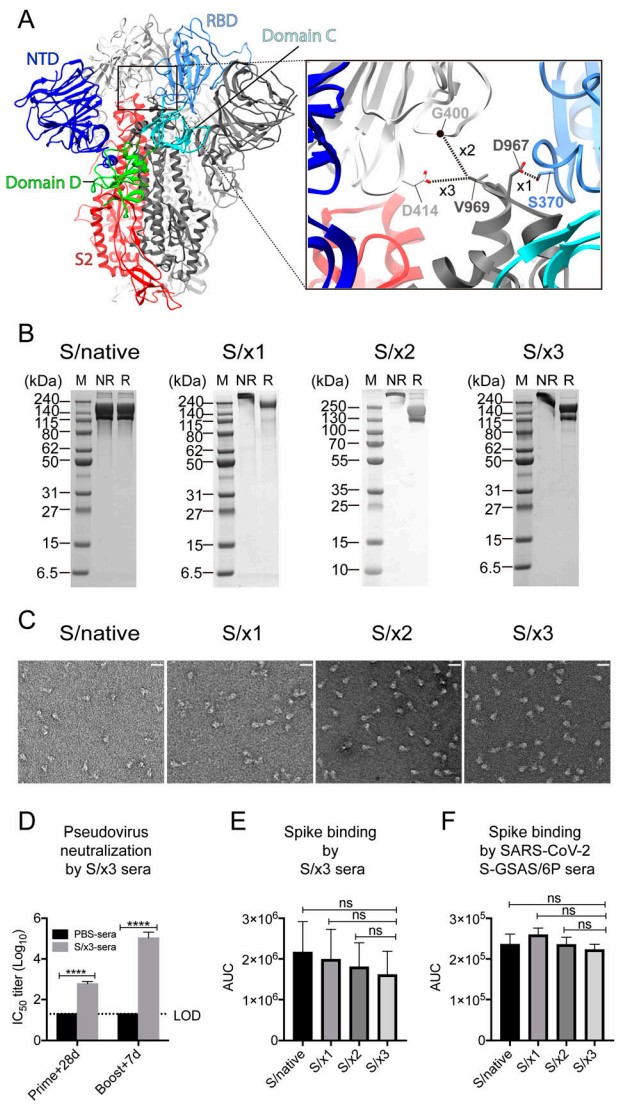

**Figure 1. Design, purification, and characterization of SARS-CoV-1 S/x1, S/x2, and S/x3 spikes.**
**(A)** Structure of SARS-CoV-1 spike (PDB: 5X58 [Yuan et al, 2017]) in closed conformation (left panel). The box indicates the location of the zoomed-in view in the right panel, engineered x1 (S370C and D967C), x2 (G400C and V969C), and x3 (D414C and V969C) disulfides are indicated in the zoomed-in view. **(B)** Coomassie-stained SDS–PAGE gels of purified SARS-CoV-1 S/native, S/x1, S/x2, and S/x3 spikes. NR or R indicates the protein sample prepared in either nonreducing or reducing condition. M indicates protein marker lanes. **(C)** Negative-stain EM images of the purified spikes. The white lines indicate a length of 250 Å. **(D, E, F)** Immunogenic and antigenic characteristics of SARS-CoV-1 spikes. **(D)** SARS-CoV-1 pseudovirus-neutralizing titers of S/x3 primed and boosted mouse immune sera. Mean and SEM (error bar) are shown, dashed line represents limit of detection. ****$P$ < 0.0001, $P$-values were analyzed by two-way ANOVA test. **(E)** Binding of S/native, S/x1, S/x2, and S/x3 spikes by the S/x3-boosted mouse immune sera. **(F)** Binding of S/native, S/x1, S/x2, and S/x3 spikes by the SARS-CoV-2 S-GSAS/6P-boosted mouse immune sera. Binding by immune sera was measured by areas under the curves derived from ELISA titration curves (Fig S3). Mean and SEM (error bar) are shown. ns, not significant, $P$-values were analyzed by one-way ANOVA test.

immunosorbent assay), the S/x3 induced immune sera exhibit binding towards SARS-CoV-1 S/native, S/x1, S/x2, and S/x3 spikes.

There are differences in reactivity of S/x3 immune sera towards different spikes, but the differences are not statistically significant (Figs 1E and S3A). SARS-CoV-2 S-GSAS/6P immune sera cross-react with S/native, S/x1, S/x2, and S/x3 spikes, and as expected, it showed reduced reactivity compared with S/x3 immune sera (Figs 1F and S3B).

## SARS-CoV-1 spikes with engineered disulfide bonds reveal additional conformations

We determined cryo-EM structures of SARS-CoV-1 S/x1, S/x2, and S/x3 spikes and we identified various previously undetected conformations for SARS-CoV-1 S through classification procedures (see Fig S4). SARS-CoV-1 S/x1 spike exhibited complex dynamics; we observed spikes adopting both symmetrical and asymmetrical conformations. Symmetrical locked-1 (three locked-1 protomers) and symmetrical locked-2 (three locked-2 protomers) S-trimers differ at Domain D-loop (residues 602–627) and FPPR structures (Figs 2A and B and 3A and B). In addition, we observed asymmetrical locked-112 (two locked-1 protomers and one locked-2 protomer, Fig 2C) and locked-122 (one locked-1 protomer and two locked-2 protomers, Fig 2D) S-trimers. Similar structural heterogeneity has been observed for the SARS-CoV-2 S-R/x3 spike in various locked conformations (Qu et al, 2022). Interestingly, SARS-CoV-1 S/x1 spike also adopts an asymmetrical closed conformation: FPPR in one protomer adopts an ordered extruding conformation, whereas being disordered in the other two protomers (Figs 2E and 3D). By cryo-EM, only symmetrical S-trimers were observed for SARS-CoV-1 S/x2 and S/x3 spikes. The S/x2 spike adopts symmetrical locked-2 (Figs 2F and 3C) and closed (Figs 2G and S5A) conformations. The S/x3 spike was only observed to adopt a symmetrical locked-1 conformation (Figs 2H and S5B). Overall, locked-1, locked-2 are structurally more rigid than closed conformations showing tighter packing and less flexibility towards RBD end of the spike (Figs S6 and S7). Structures within each identified conformation categories, namely locked-1, locked-2, and closed, share almost identical structural features. Between different conformation categories, there are variations in structures of Domain D, FPPR, and positioning of RBD. These variations appear to result in a different S-trimer packing (Fig S7). Distinct structures from different conformation categories are summarized and described in detail in the next sections.

## Structural features of SARS-CoV-1 spikes in different conformations

To further understand conformational dynamics of SARS-CoV-1 spike, representative structures in each conformation category are summarized in Fig 3 with features highlighted (additional structures are shown in Fig S5). In SARS-CoV-1 S/x1 locked-1 protomers (Fig 3A), a linoleic acid can be modelled in the density identified in a lipid-binding pocket within the RBD. This pocket was previously identified in SARS-CoV-2 spike and it has been proposed to be a conserved feature in betacoronavirus spikes (Toelzer et al, 2020). By cryo-EM, we confirmed that the lipid is bound in a highly conserved fashion between SARS-CoV-1 and SARS-CoV-2 spikes. The aliphatic tail of the bound lipid is surrounded by aromatic

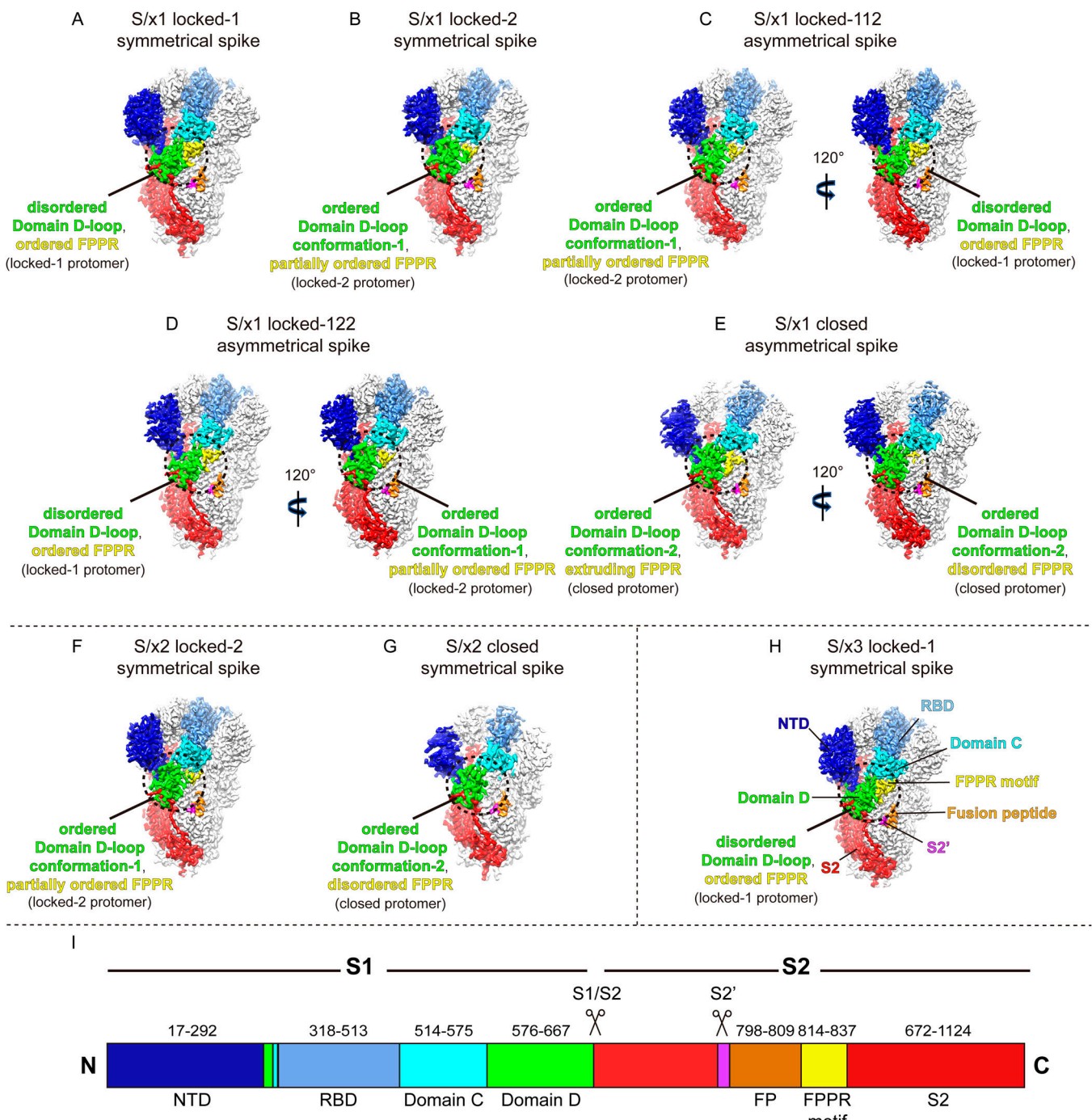

**Figure 2. Cryo-EM structures of SARS-CoV-1 S/x1, S/x2, and S/x3 spikes reveal multiple spike conformations.**
**(A, B, C, D, E, F, G, H)** Cryo-EM densities are shown for structures determined for SARS-CoV-1 S/x1, S/x2, and S/x3 spikes. Featured protomers within the S-trimer densities are colored. S1 structural domains—NTD, RBD, Domain C, and Domain D are colored blue, light-blue, cyan, and green, respectively. S2 is colored red with S2 structural elements—S2′, fusion peptide (FP), and fusion peptide proximal region (FPPR) highlighted magenta, orange, and yellow, respectively. **(A, B)** Symmetrical locked-1 and locked-2 conformations identified for S/x1. S/x1 locked-1 conformation has a disordered Domain D-loop and ordered FPPR, whereas Domain D-loop in S/x1 locked-2 conformation adopts an ordered conformation (Domain D-loop conformation-1) and FPPR is partially ordered. Domain D regions where major structural rearrangement occurs are highlighted within the dashed circles. **(C, D)** Asymmetrical locked-112 and locked-211 conformations identified for S/x1 spike. S-trimers in these conformations are assembled from different combinations of locked-1 and locked-2 protomers. S-trimers are rotated by 120° to show protomers of different conformations. **(E)** S/x1 spike in an asymmetrical closed conformation. The spike is rotated 120° to show structural differences in FPPR between protomers. **(F, G, H)** Symmetrical conformations identified for S/x2 and S/x3 spikes. **(I)** A schematic drawing of SARS-CoV-1 spike sequence showing its structural domains and elements.

## A  S/x1 locked-1

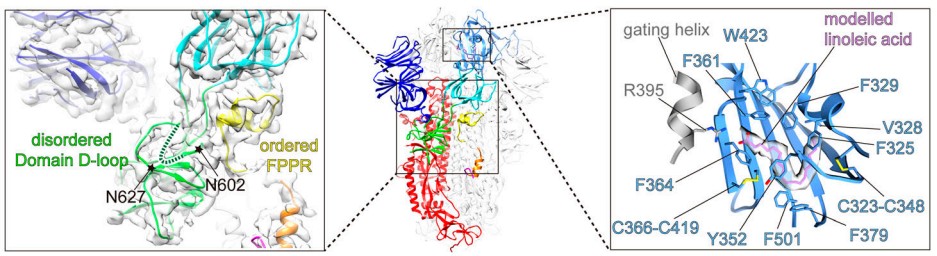

## B  S/x1 locked-2

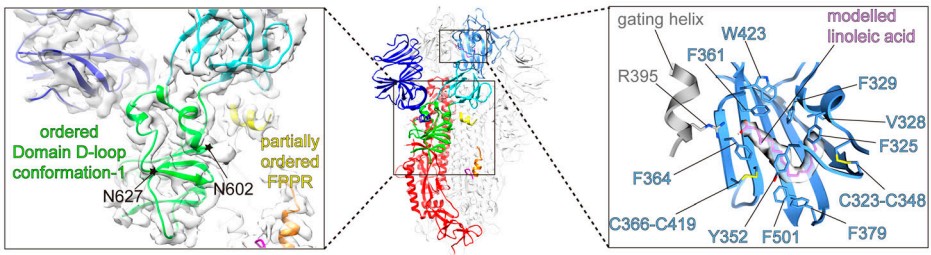

## C  S/x2 locked-2

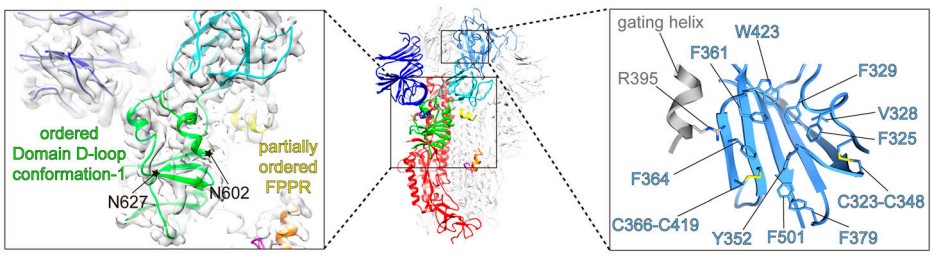

## D  S/x1 closed-extruding FPPR

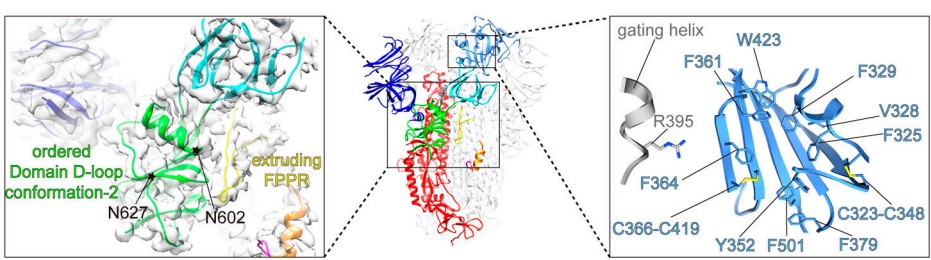

## E  S/x1 closed-disordered FPPR

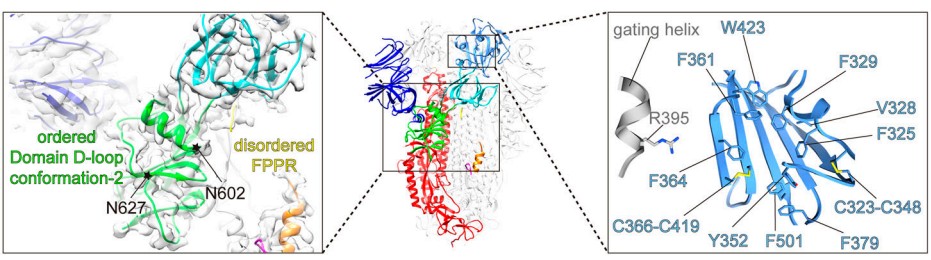

**Figure 3.  Structural features of spike protein protomers in different conformations.**

**(A, B, C, D, E)** Representative protomers of different conformation categories and with distinctive features were summarized from structures shown in Fig 2. Molecular models are shown in the middle panels. Left panels show cryo-EM densities of Domain D and the surrounding regions with fitted molecular models. Flexible regions (Domain D-loop) between N602–N627 where structural rearrangements occur are highlighted and indicated by star symbols. Structural features in each structure are shown and indicated. Right panels show lipid-binding pockets within SARS-CoV-1 RBDs of different conformations. Hydrophobic amino acid sidechains forming the lipid-binding pocket are shown in stick representations. Gating helices from the neighboring RBDs are shown in grey with the head group-interacting R395 in stick representations. **(A, B)** Lipid-binding pockets in S/x1 locked-1 (A) and locked-2 (B) conformations show clear lipid densities with modelled lipids in purple stick representations. **(C, D, E)** Lipid-binding pockets appear to be unoccupied in S/x2 locked-2 (C) and S/x1 closed (D, E) conformations showing the absence of lipid density.

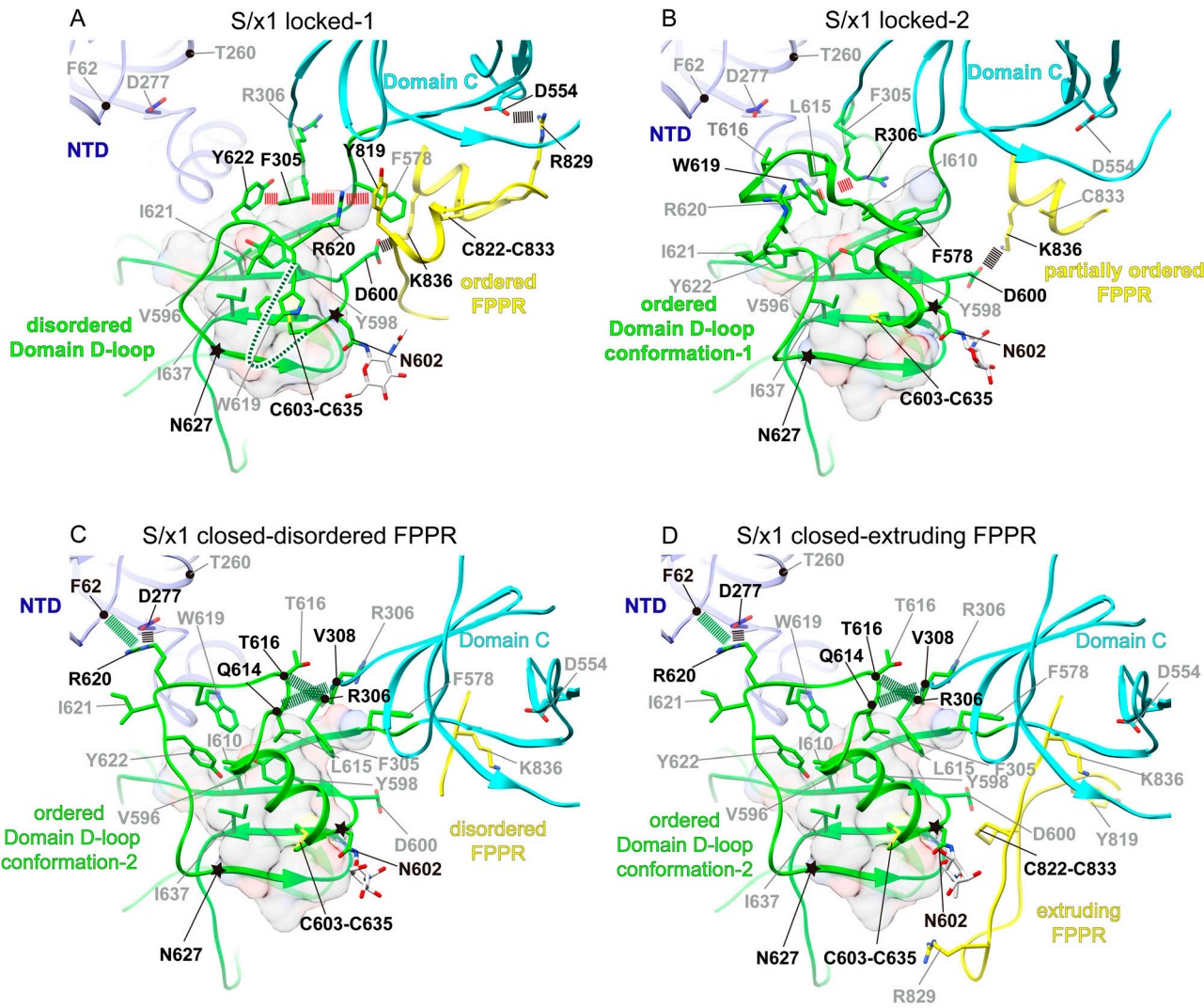

**Figure 4. Structural rearrangements in Domain D among SARS-CoV-1 spikes of different conformations.**
**(A)** Domain D and the surrounding structural elements in locked-1 conformation. Residues N602–W619 within the Domain D-loop are disordered and represented by a green dashed line. Domain D-loop (residues N602–N627) where structural rearrangements occur between conformations is highlighted by star symbols. Cation–π and π–π interactions in the interdigitation interaction involving Y622, F305, R620, and Y819 are indicated by red dashed lines. Salt bridges are indicated by black dashed lines. Residues involved in featured interactions are highlighted by black labels. Hydrophobic residues (V596, Y598, and I637) within Domain D β-sheets are shown along with a transparent molecular surface of the Domain D hydrophobic core. A folded and ordered FPPR (yellow) is identified in locked-1 conformation. **(B)** Domain D-loop in locked-2 conformation refolds to form ordered structures. A new cation–π interaction (red dashed line) is formed between R306 and W619. FPPR is partially disordered, and the interaction between D600 of Domain D and K836 of FPPR is retained (black dashed line). **(C, D)** Further refolding of the Domain D-loop is observed in closed conformations and hydrophobic residues within the loop pack differently against the Domain D hydrophobic core. Interactions (black and green dashed lines) are formed to stabilize the folded Domain D-loop. **(D)** FPPRs are normally disordered in spikes of closed conformation, and an ordered extruding FPPR is observed for one protomer of the closed S/x1 spike.

amino acids lining the pocket. The pocket is gated by a small helix from a neighboring protomer to allow R395 within the gating helix to form a salt bridge to the negatively charged lipid head group (Fig 3A, right panel). Therefore, the bound lipid is interacting with RBDs of two neighboring protomers and these interactions are consistent with the proposal that they stabilize locked conformations (Toelzer et al, 2020). In the S/x1 locked-1 protomer, cryo-EM density for residues between 602 and 619 is not visible suggesting that these residues form a flexible loop (Fig 3A, left panel and Fig 4A). In an unusual dimer of locked-1 conformation SARS-CoV-2

spikes, formation of a large extended loop by this region was confirmed (Bangaru et al, 2020). In S/x1 locked-2 spike protomers (Fig 3B), linoleic acid is bound in the RBD in an almost identical way as observed for the locked-1 protomer (Fig 3B, right panel). Different from locked-1 spike protomers, residues between 602 and 619 in S/x1 locked-2 spike protomers refold into ordered structures showing clear cryo-EM density (Fig 3B, left panel and Fig 4B). The S/x2 locked-2 spike protomer displays cryo-EM density showing an ordered 602–619 region in Domain D (Fig 3C, left panel), highly similar to S/x1 locked-2 spike protomer (Fig 3B, left panel).

Surprisingly, the lipid-binding pocket is unoccupied even though the entrance to the pocket is closed by the gating helix (Fig 3C, right panel). The S/x3 spike was only observed to adopt a symmetrical locked-1 conformation, its protomers share very similar features as S/x1 locked-1 protomers, with bound lipid and residues 602–619 forming a flexible loop (Fig S5B).

We identified clear densities consistent with biliverdin molecules in S/x1 locked-1, locked-2, locked-112, and locked-122 structures in a NTD pocket equivalent to the previously identified biliverdin-binding pocket in SARS-CoV-2 S (Rosa et al, 2021; Qu et al, 2022) (Fig S5C). Despite imaged under similar conditions, no density or very weak densities for biliverdin were observed in the biliverdin-binding pockets of the other currently reported structures. In contrast, biliverdin densities were evident in spikes of locked and closed conformations for SARS-CoV-2 (Qu et al, 2022). We noticed certain degrees of sequence conservation within the largely hydrophobic biliverdin-binding pocket between SARS-CoV-1 and SARS-CoV-2 S, but there are changes at several interacting residues (Fig S5C). Molecular simulations suggested sequence changes in this pocket could affect biliverdin binding (Qu et al, 2021). It has been suggested that biliverdin binding could interfere with antibody binding (Cerutti et al, 2021; Rosa et al, 2021). Spike NTD has been previously observed to bind cofactors with unknown functions with regard to the CoV life cycle, for example, folic acid has been identified to bind MERS-CoV spike NTD (Pallesen et al, 2017). SARS-CoV-1 and SARS-CoV-2 represent two distinct lineages of Sarbecoviruses (Guo et al, 2021), conservation of biliverdin-binding pocket between their spikes, may suggest an unidentified physiological role for this molecule in the Sarbecovirus infection cycle.

Two distinct conformations were identified for protomers in the closed S/x1 asymmetrical spike (Figs 3D and E). In both conformations, lipid-binding pockets were unoccupied, consistent with the fact that the gating helices from neighboring protomers are dislocated leaving lipid-binding pockets open (Figs 3D and E, right panels). In both closed protomers, clear cryo-EM densities were observed for residues between 602 and 619 (Figs 3D and E, left panels and Figs 4C and D). However, they adopted a different structure comparing with the equivalent region in the locked-2 protomers (compared with Figs 3B and C, left panels). FPPR has been found disordered in closed SARS-CoV-2 spikes, in line with this observation, FPPR is disordered in two of the three protomers within the SARS-CoV-1 S/x1 asymmetrical closed spike. However, surprisingly, one of the protomers has an FPPR forming an ordered extruding structure (Fig 3D, left panel). The S/x2 spike can adopt a symmetrical closed conformation, spike protomer of this conformation has an empty lipid-binding pocket and disordered FPPR (Fig S5A). Therefore, S/x2 closed protomer has almost identical features to the S/x1 closed protomer with a disordered FPPR (compare Figs 3E and S5A).

### Domain D adopts different structures in SARS-CoV-1 spikes of different conformations

We summarized four representative structural arrangements around Domain D identified from determined cryo-EM structures. We examined interactions around Domain D in these structures in

details. In locked-1 protomers, interdigitation interactions between Y622, F305, R620, and Y819 are formed through π–π and cation–π interactions (Fig 4A). These interactions connect the structural elements of Domain D (Y622, R620), Domain C–D hinge region (F305), and FPPR (Y819) (Fig 4A). Equivalent interdigitation interactions were observed in SARS-CoV-2 locked-1 spike. We have proposed that these interactions rigidify this area preventing structural transitions needed for RBD opening (Qu et al, 2022). Density for residues 602–619 is not visible and the equivalent region is known to form a large disordered loop in SARS-CoV-2 S adopting a locked-1 conformation (Bangaru et al, 2020) (Fig 4A). Hydrophobic residues W619, I621 located at the C-terminus of the loop interact with V596, Y598, and I637 of the Domain D hydrophobic core (Fig 4A). Both the interdigitation and hydrophobic interactions appear to hold the disordered loop in place. A salt bridge is formed between D600 of Domain D and K836 of FPPR (Fig 4A). D600 is equivalent to D614 in SARS-CoV-2 spike and D614G is the earliest mutation which became fixed in SARS-CoV-2 spike since the pandemic and all subsequent variants bear this mutation (Grubaugh et al, 2020; Korber et al, 2020). Likely stabilized by the participated interactions, FPPR is ordered in locked-1 conformation consisting of two small helices connected by a disulfide between C822–C833 (Fig 4A). The presence of this structural motif is known to be incompatible with RBD in the "up" position (Xiong et al, 2020).

In the locked-2 conformation, the highly dynamic region between residues 602–627 in Domain D (Domain D-loop) undergoes drastic refolding (Fig 4B) to adopt a fully ordered "Domain D-loop conformation-1" with several mini-helices within the loop. Interdigitation interactions connecting Domain D, Domain C–D hinge region, and FPPR in the locked-1 conformation are broken. Residues between 602 and 619 become ordered in the locked-2 conformation (Fig 4B). A change of Domain C–D hinge region structure allows residues R306 and F578 to interact with W619 and the Domain D hydrophobic core by cation–π and hydrophobic interactions, respectively. These interactions likely still rigidify this region to maintain the more tightly packed locked-2 conformation (Figs S6 and S7). Most interactions between FPPR and other parts of spike, including the interdigitation interactions are lost; only the salt bridge between K836 of FPPR and D600 of Domain D is retained. FPPR is partially ordered with clear density only for the helix formed by residues 830–838 (Figs 3B and C and 4B).

Further structural changes around Domain D occur in the S/x1 closed conformations, a large structural change caused bending and rotation of Domain C–D hinge region allowing residue F305 to interact with the Domain D hydrophobic core (Figs 4C and D). The large Domain D-loop (residues 602–627) assumes a different ordered conformation (here, we refer it as "Domain D-loop conformation-2"), the three mini-helices in Domain D-loop of locked-2 conformation unfold and a longer helix between 602 and 610 is formed (Figs 4C and D). This conformational change allowed R620 to interact with NTD F62 mainchain carbonyl and D277 sidechain carboxylate by hydrogen bond and salt bridge, respectively (Fig 4C and D). Because of this refolding, W619 and Y622 interact with the Domain D hydrophobic core differently (Fig 4C and D). Although the above described interactions involve residues conserved between SARS-CoV-1 and SARS-CoV-2 spikes, cryo-EM density was not observed for Domain D-loop in

A  S/x1 locked-1 (colored)
   S/x1 locked-2 (grey)

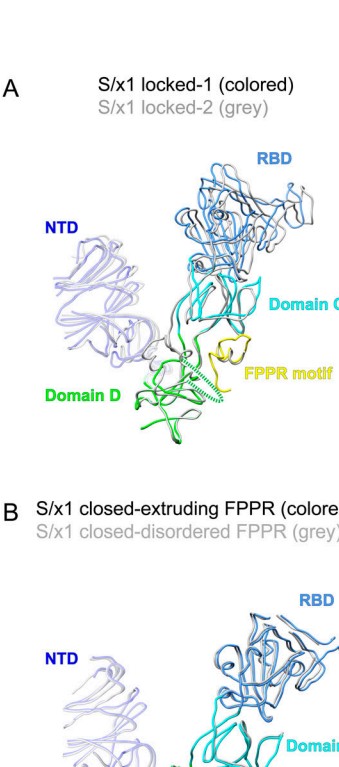

B  S/x1 closed-extruding FPPR (colored)
   S/x1 closed-disordered FPPR (grey)

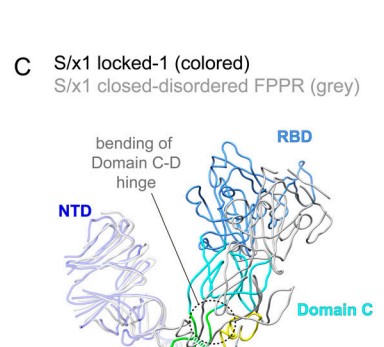

C  S/x1 locked-1 (colored)
   S/x1 closed-disordered FPPR (grey)

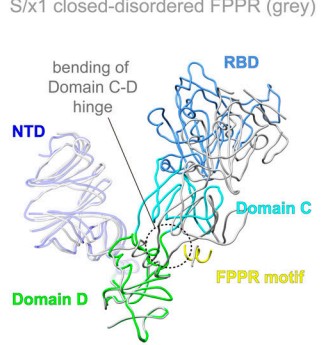

D  S/x1 locked-2 (colored)
   S/x1 closed-disordered FPPR (grey)

E  S/x1 locked-1 (colored)
   SARS-CoV-2 S locked-1 (7XTZ, grey)

F  S/x1 locked-2 (colored)
   SARS-CoV-2 S locked-2 (7XU2, grey)

G  S/x1 closed-disordered FPPR (colored)
   SARS-CoV-2 S closed (7XU3, grey)

H  S/x1 locked-2 (colored)
   pH 5.5 SARS-CoV-2 S  (6XM5, grey)

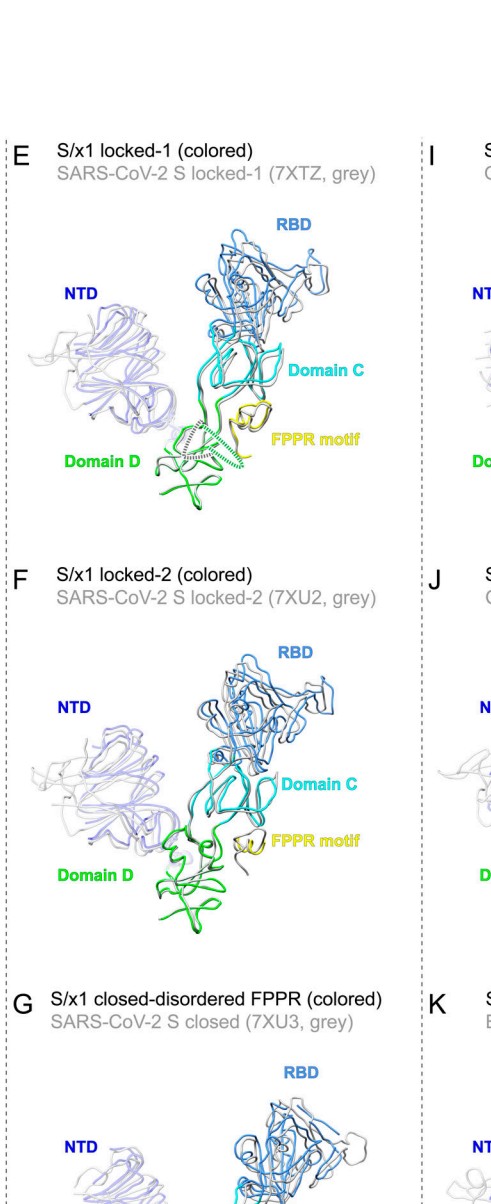

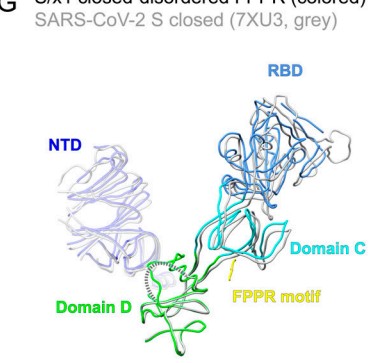

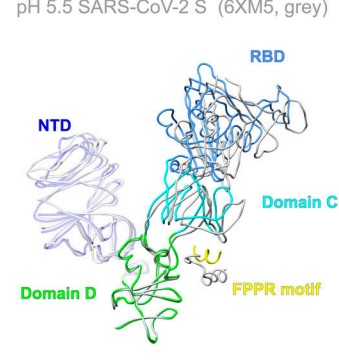

I  S/x1 locked-2 (colored)
   GD Pangolin-CoV S (7BBH, grey)

J  S/x1 locked-2 (colored)
   GX Pangolin-CoV S (7CN8, grey)

K  S/x1 locked-2 (colored)
   Bat-CoV RaTG13 S (7CN4, grey)

L  S/x1 locked-2 (colored)
   Bat-CoV RaTG13 S (6ZGF, grey)

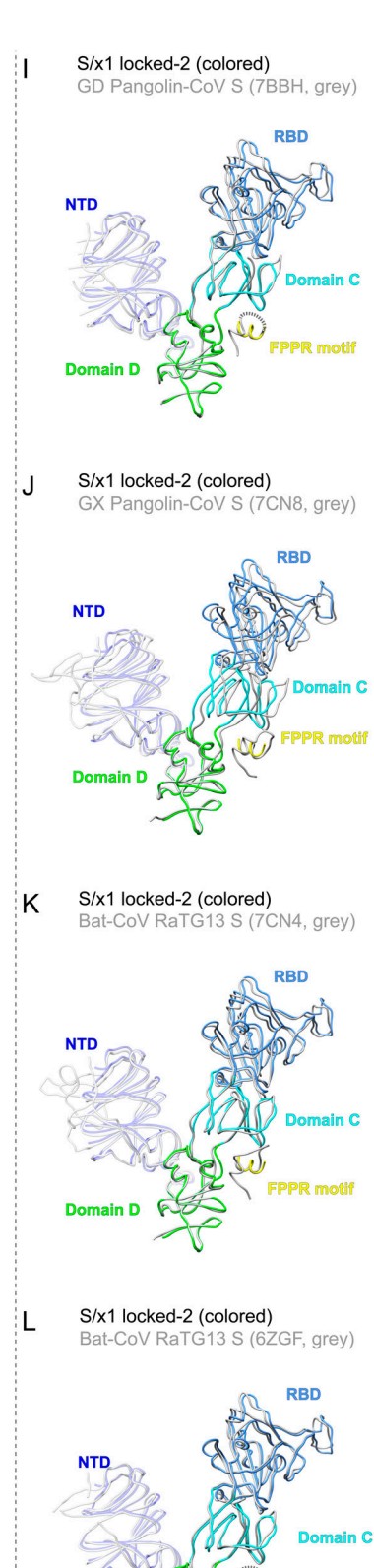

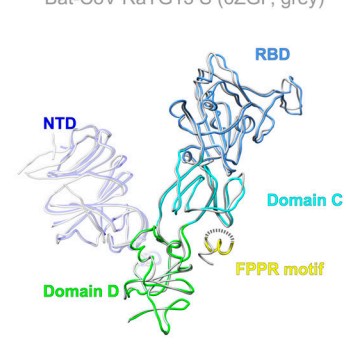

SARS-CoV-2 spike of closed conformation, suggesting that the equivalent Domain D-loop adopts a different flexible conformation (Qu et al, 2022). We identified numerous residues in or around Domain D differ between SARS-CoV-1 and SARS-CoV-2 spikes (Fig S8) making it difficult to pin down which amino acid changes result in different Domain D structural dynamics between the two spikes. We noted though, if T260 in SARS-CoV-1 S is substituted for an arginine found in the equivalent position of SARS-CoV-2 S (Figs 4C and D and S8), it would be in close proximity to R620 of Domain D-loop, repulsion between the two positively charged residues would be unfavorable for Domain D-loop to adopt the ordered "Domain D-loop conformation-2" that is featured in closed conformations of SARS-CoV-1 spike.

### SARS-CoV-1 and SARS-CoV-2 spikes sample similar conformations

Previously, we found domain movement in a protomer alters trimer packing resulting in differences in the spike quaternary structure between locked and closed conformations (Qu et al, 2022). To further understand conformational dynamics of the SARS-CoV-1 spike, S1 of SARS-CoV-1 spikes of different conformations were structurally aligned using Domain D as the reference (Figs 5 and S9A–L). The overlaps show that drastic refolding of Domain D between locked-1 and locked-2 conformations alters the structure of the Domain C–D hinge region and affects the structure of FPPR (Fig 5A). Despite these extensive structural changes, positioning of NTD and RBD is largely unaffected, and only small shifts in NTD and RBD positions are observed (Fig 5A). Similarly, the two closed conformations are almost superimposable when aligned, despite structural differences in FPPR (Fig 5B). In contrast, comparison of SARS-CoV-1 S locked-1 and locked-2 conformations to closed conformations revealed that positioning of RBD differ considerably between locked and closed conformations (Fig 5C and D). The overlap of locked and closed conformations revealed that RBD movement originates from the bendings in the Domain C–D hinge region (Fig 5C and D). We previously also identified similar characteristics for SARS-CoV-2 S, structural transition from locked conformations to closed conformation is associated with refolding of Domain D, bending of Domain C-D hinge region, and movement of RBD (Qu et al, 2022). Overlaps of SARS-CoV-1 and SARS-CoV-2 S1 of different conformations reveal that spikes of the two CoVs share very similar locked-1, locked-2, and closed conformations (Fig 5E–G).

Several details are different: first, whereas FPPR is fully ordered in SARS-CoV-2 S of both locked-1 and locked-2 conformations (Fig 5E and F), FPPR is only fully ordered in SARS-CoV-1 S of locked-1 conformation (Fig 5E). Second, SARS-CoV-2 S in closed conformation is characterized by its disordered Domain D-loop and FPPR (Fig 5G), whereas in SARS-CoV-1 S, Domain D-loop is refolded to adopt an ordered structure but distinct from that in locked-2 conformation (Figs 4B–D and 5G). Third, we identified an asymmetrical closed conformation for SARS-CoV-1 S in which FPPR is dynamic adopting either the extruding loop or disordered structures, whereas FPPR is usually fully disordered in closed conformation SARS-CoV-2 S (Fig 5B). Previously, dynamics was observed for FPPR when SARS-CoV-2 spike was treated with acidic pH (Fig 5H) (Zhou et al, 2020).

## Discussion

Class I viral membrane fusion glycoproteins are evolved to have metastable prefusion conformation (Harrison, 2008). Introduction of disulfides into viral fusogenic glycoproteins has been proven useful to stabilize flu HA (Godley et al, 1992; Lee et al, 2015), HIV GP160 (Sanders et al, 2013), and paramyxovirus F proteins (McLellan et al, 2013; Wong et al, 2016) in various prefusion conformations for mechanistic, structural, and vaccinology studies. Here, we introduced x1, x2, x3 disulfides previously designed for SARS-CoV-2 S into SARS-CoV-1 S; this allowed structural characterization of previously unobserved conformations for the SARS-CoV-1 spike. Newly determined structures revealed that SARS-CoV-1 S is able to form symmetrical locked-1, locked-2 trimers or asymmetrical trimers made up by locked-1 and locked-2 protomers in various combinations. An asymmetrical closed trimer was also observed with one of the protomer featuring an extruding FPPR. Therefore, together with previously determined structures, it is confirmed that the SARS-CoV-1 spike can adopt locked-1, locked-2, closed, and RBD "up" open conformations. SARS-CoV-1 and SARS-CoV-2 spikes appear to sample similar conformations and they both exhibit complex conformational dynamics.

Structures of a few other Sarbecovirus spikes have also been determined. SARSr-CoV spikes, namely Guangdong (GD) pangolin-CoV S (Wrobel et al, 2021), Guangxi (GX) pangolin-CoV S (Zhang et al, 2021b), and RaTG13 bat-CoV (Wrobel et al, 2020; Zhang et al, 2021b) S, only adopt the locked-2 conformation when imaged by cryo-EM (Fig 5I–L). Therefore, SARSr-CoV spikes are different from SARS-CoV-1 and SARS-CoV-2 spikes exhibiting reduced structural dynamics. Alternatively, closed/open conformations of SARSr-CoV spikes may not be stable enough to withstand cryo-EM sample preparation procedures and that the more structurally rigid spikes in locked conformations were preferentially imaged.

For SARS-CoV-2 (Toelzer et al, 2020) and SARSr-CoV spikes (Zhang et al, 2021b), locked conformations are often associated with lipid binding in RBD and lipid binding has been proposed to stabilize

**Figure 5. SARS-CoV-1 and SARS-CoV-2 spikes are highly dynamic adopting multiple shared conformations, other SARSr-CoV spikes exhibit reduced dynamics.** **(A, B, C, D)** Comparison of SARS-CoV-1 spike S1 portions of different conformations. Structures are shown in ribbon representations. Positions of S1 structural domains remain largely unchanged between locked-1 and locked-2 conformations despite structural rearrangements in Domain D. **(B)** An overlap of two S/x1 closed conformations reveals highly similar positions of S1 structural domains. **(C, D)** Overlaps of locked-1 and locked-2 conformations to closed conformation reveal a large motion of RBD between locked and closed conformations originating from the bendings in the Domain C–D hinge region. **(E, F, G, H)** Overlaps of SARS-CoV-1 and SARS-CoV-2 spike S1 structures of related conformations to show similarities (PDBs: 7XTZ, 7XU2, 7XU3 [Qu et al, 2022]; 6XM5 [Zhou et al, 2020]). **(I, J, K, L)** Overlaps of SARS-CoV-1 and SARSr-CoV spike S1 structures of related conformations to identify similarities (PDBs: 7BBH [Wrobel et al, 2021]; 7CN8, 7CN4 [Zhang et al, 2021b]; 6ZGF [Wrobel et al, 2020]).

locked S-trimers (Toelzer et al, 2020). In this study, association of lipid binding and locked conformations were observed for SARS-CoV-1 S/x1 and S/x3 spikes. Previously, we and others found that low pH was able to convert a closed/open SARS-CoV-2 spike to locked conformations (Zhou et al, 2020; Qu et al, 2022). Based on these observations and the egress pathway characterized for beta-coronaviruses (Ghosh et al, 2020), we have proposed that locked conformations are associated with virus particle assembly in low-pH, lipid-rich endomembrane structures (Qu et al, 2022).

In this study, we confirmed that locked-1 and locked-2 conformations are shared between SARS-CoV-1 and SARS-CoV-2 spikes and locked-2 conformations are shared among all characterized Sarbecovirus spikes. Structural characterization reveals that, although details differ between locked-1 and locked-2 conformations, restraining RBD movement by interactions between Domain D and Domain C–D hinge region is shared by locked conformations of Sarbecovirus spikes. Conservation of RBD-restraining mechanisms in locked conformations among Sarbecovirus spikes suggests S-trimers in locked conformations may play a conserved function in the Sarbecovirus life cycle.

Previously, only closed and open conformations were observed for purified SARS-CoV-1 S (Fig S9G–L). We have attempted to convert native SARS-CoV-1 S ectodomain (S/native) without engineered disulfide to locked conformation by prolonged incubation under low pH (pH 5.5 for 24 h); however, we only observed open and closed spikes after incubation (Fig S4). In line with previously determined SARS-CoV-1 S structures (Gui et al, 2017; Yuan et al, 2017), Domain D-loops in closed and open spike structures determined under low pH are ordered, adopting "Domain D-loop conformation-2," showing almost identical conformations as previously detailed for the closed conformation (Figs S9 and S10). Of note, during preparation of this manuscript, a subpopulation of purified insect cells expressed (presumably expressed in low-pH insect cell media) SARS-CoV-1 spike ectodomain adopting a locked-2 conformation with lipid bound in RBD was reported (Toelzer et al, 2022). We speculate that unlike SARS-CoV-2 S, where Domain D-loops are disordered in closed and open conformations, breaking interactions involving the ordered Domain D-loops in closed and open conformations represents a barrier to revert closed/open spike to locked conformations for SARS-CoV-1 S. This difference reflects how changes in Domain D may affect the dynamics of Sarbecovirus spikes. Indeed, D614G mutation (Yurkovetskiy et al, 2020; Zhang et al, 2021a) and proteolytic cleavage by cathepsin L (Zhao et al, 2022), both within Domain D, are known to affect the dynamics of the SARS-CoV-2 spike.

In summary, SARS-CoV-1 and SARS-CoV-2 spikes exhibit complex conformational dynamics among Sarbecovirus spikes. Biological functions of these different conformations remain to be further confirmed or elucidated. In addition, the underlying mechanisms by which different Sarbecovirus spikes sample various conformations differently and the associated biological consequences remain poorly understood. Future "in situ" structural studies under more conditions may provide further clues for these questions. Nevertheless, in this study, we demonstrated that introduction of stabilizing disulfides in the SARS-CoV-1 spike is useful for capturing otherwise transient conformations allowing more thorough characterization of spike dynamics. We further demonstrated an engineered RBD "down" S/x3 spike as an immunogen. A survey of structurally characterized SARS-CoV-1 S binding mAbs identify that only 2 out 19 of the structurally characterized SARS-CoV-1 S binding mAbs are able to bind RBD "down" S-trimer without clash (Fig S11). Together with previously determined SARS-CoV-2 spike structures, we believe that the conserved features characterized for SARS-CoV-1 and SARS-CoV-2 spikes should help further understanding of Sarbecovirus spike function.

## Materials and Methods

### Protein expression and purification

S/native construct was generated by cloning S gene of SARS-CoV-1 (SARS coronavirus Tor2, NCBI: txid227984) between amino acid residues 15–1,193 into a modified pcDNA3.1 vector previously described (Walls et al, 2019; Xiong et al, 2020). The S/native construct encodes an exogenous N-terminal signal peptide MGILPSPGMPALLSLVSLLSVLLMGC-VAETGT derived from μ-phosphatase and a C-terminal extension GSGR*ENLYFQ*GGGGGSGYIPEAPRDGQAYVRKDGEWVLLSTFLGHHHHHH. The C-terminal extension contains a tobacco etch virus (TEV) protease cleavage site (italic), a T4 trimerization foldon (underlined), and a hexa-histidine tag (bold). This construct is called S/native. S/x1 (S370C and D967C), S/x2 (G400C and V969C), and S/x3 (D414C and V969C) containing the introduced cysteines were generated by PCR.

Spike proteins were expressed by transient transfection in Expi293 cells; 1 mg of DNA was transfected into 1 liter of cells using PEI transfection reagent. After transfection, the cells were cultured at 33°C for 5 d. Supernatants were harvested by centrifugation and 25 mM phosphate, pH 8.0, 5 mM imidazole, 300 mM NaCl were supplemented. The supernatants were recirculated onto a 5-ml Talon Cobalt column for ~2.5 times and the column was washed by 100 ml buffer A (25 mM phosphate pH 8.0, 5 mM imidazole, 300 mM NaCl). Spike protein was eluted by a linear gradient using increasing concentration of Buffer B (25 mM phosphate pH 8.0, 500 mM imidazole, 300 mM NaCl). Purified spike proteins were quality checked by SDS–PAGE and buffered and exchanged into PBS and frozen at –80°C until further use.

### Negative-staining EM

Spike proteins were diluted to 0.06 mg/ml and 3 μl samples were applied onto glow-discharged (15 mA, 45 s glow), carbon-coated copper grids. Protein samples were absorbed for 1 min before grids were washed with water once. The samples were stained twice with 0.75% (wt/vol) uranyl formate solution. Grids were air-dried and imaged in an FEI Tecnai G2 Spirit transmission electron microscope operating at 120 keV equipped with a CCD camera.

### Disulfide bond reduction under native conditions

S/x1, S/x2, and S/x3 spike proteins were incubated with 0, 2.5, 5, 10, and 20 mM DTT for 5 or 60 min at room temperature in PBS. Reactions were quenched by addition of 55 mM iodoacetic acid for 10 min in the dark at room temperature. Reaction mixtures were

mixed with 4× nonreducing loading buffer and were analysed by SDS–PAGE.

## Animal immunization

Animal study was approved by the Ethics Committee of the First Affiliated Hospital of Guangzhou Medical University (IACUC number: 2020164). 6–8-wk-old BALB/c females were immunized intramuscularly with 10 μg of purified protein mixed with 100 μg of Sigma Adjuvant System in a total volume of 100 μl of PBS. After a 4-wk interval, mice were boosted with the same antigen in the same formulation. Day 28 after the priming immunization and day 7 after boost immunization, sera were collected from immunized mice to detect the antigen-specific humoral response by ELISA and pseudovirus neutralization assay.

## Pseudovirus neutralization assay

Pseudotyped lentiviruses were produced in 293T cells as previously described (Feng et al, 2020) by co-transfecting a plasmid expressing SARS-CoV-1 S protein, a packaging vector, and a reporter vector carrying an expression cassette of firefly luciferase. The 4× serially diluted heat-inactivated (HI) serum (56°C for 30 min) were incubated with the SARS-CoV-1 pseudotyped virus at 37°C for 1 h. The mixture was subsequently incubated with HeLa-ACE2 cells for 72 h. The cells were washed twice with PBS and lysed with lysis buffer before measuring luciferase activity. The neutralization titers were calculated as antibody dilutions at which the luciferase activity was reduced to 50% of that from the virus-only wells.

## ELISA assay of antiserum binding to SARS-CoV-1 spikes

96-well assay plates were coated with 100 μl per well of SARS-CoV-1 S/native, S/x1, S/x2 or S/x3 spike proteins at 1 μg/ml in PBS overnight at 4°C. After standard washing, 10% fetal calf serum (FBS, 200 μl per well) was added to block for 2 h at 37°C. After washing the wells three times with PBS-0.1% Tween 20 (MP Biomedicals), 100 μl semilogarithmic dilutions of antisera in PBS was added to each well and incubated for 2 h at 37°C. After washing the wells three times with PBS-0.1% Tween 20, the plates were incubated with 1:10,000 dilutions of HRP-labelled goat anti-mouse IgG (H + L) (Jackson ImmunoResearch Laboratories) in 10% FBS for 1 h at 37°C. After washing the wells six times with PBS-0.1% Tween 20, 100 μl per well of TMB (3,3′,5,5′-tetramethylbenzidine) solution (Merck Millipore) was added and developed for 10 min at room temperature. Reactions were stopped by adding 50 μl 2 M sulphuric acid and OD values at 450 nm were measured in a plate reader.

## Cryo-EM sample preparation

Immediately before cryo-EM sample preparation, spike samples in PBS (S/x1 at 4.7 mg/ml, S/x2 at 6.5 mg/ml, S/x3 at 2.4 mg/ml, S/native at 6.68 mg/ml) were thawed and TEV protease (2.5 mg/ml in PBS) was added to a 10:1 spike/TEV protease w/w ratio. The mixtures were incubated at room temperature for 1 h before using for grid preparation. For the low-pH S/native spike sample, after the

TEV protease incubation, per 9 μl of the protein solution was incubated with 1 μl of 1 M pH 5.5 sodium acetate buffer (giving a final pH of 5.54 for the protein solution) overnight at room temperature. Quantifoil R1.2/1.3 copper grids were glow-discharged (15 mA, 30 s) before being used for specimen preparation. 3 μl protein solution was mixed with 0.3 μl 1% octyl-glucoside (final concentrations of OG were 0.1%) immediately before being applied to grid. Grids were blotted (3 s, force 4) and rapidly plunge-frozen in liquid ethane using a Vitrobot (Thermo Fisher Scientific) in 100% humidity at 22°C. Grids were screened in a Talos Arctica G2 electron microscope (Thermo Fisher Scientific) operated by EM facility of Guangzhou Institutes of Biomedicine and Health. Suitable grids with clear side- and top-views of spikes were identified and datasets were collected in the Talos Arctica G2 electron microscope, or a Titan Krios G3i electron microscope (Thermo Fisher Scientific) operated by EM center of Southern University of Science and Technology (SUSTech), Shenzhen.

## Cryo-EM data collection

The Talos Arctica electron microscope was operating at 200 keV and Serial EM software v3.8.7 was used for data collection. On a K3 direct detection camera (Gatan), micrographs were acquired at a nominal magnification of 45,000×, with a calibrated pixel size of 0.88 Å and a defocus range of −0.8 to −2.5 μm. For the S/x2 and S/x3 samples, each movie was exposed for 1.6 s with a dose rate of 30 e$^-$/pixel/second fractionated into 27 frames, resulting in a total dose of 62 e$^-$/Å$^2$. For the low-pH S/native sample, each movie was exposed for 1.8 s with a dose rate of 26 e$^-$/pixel/second fractionated into 27 frames, resulting in a total dose of 60 e$^-$/Å$^2$.

The Titan Krios G3i electron microscope was operating at 300 keV and EPU software (Thermo Fisher Scientific) was used for data collection. On a BioQuantum K3 direct detection camera (Gatan) using a 20 eV filter slit width operated in zero-loss mode, micrographs were recorded at a nominal magnification of 81,000×, with a calibrated pixel size of 1.095 Å and a defocus range of −0.8 to −2.0 μm. Each movie was exposed for 2.37 s with a dose rate of 25 e$^-$/pixel/second, fractionated into 38 frames, resulting in a total dose of 50 e$^-$/Å$^2$.

## Cryo-EM data processing

Movies were aligned using a MotionCor2-like algorithm (Zheng et al, 2017) implemented in RELION v3.1 (Zivanov et al, 2020), Contrast transfer function-estimation and non-templated particle picking were carried out by Warp v1.09 (Tegunov & Cramer, 2019); 4,604,239 particles were picked in the S/x1 dataset, 967,643 particles were picked in the S/x2 dataset, 82,693 particles were picked in the S/x3 dataset, and 1,375,445 particles were picked in the S/native low-pH dataset. Extracted particles were imported back into RELION. An EM map of SARS-CoV-2 S-R/x2 spike (EMD-11329) (Xiong et al, 2020) in closed conformation was filtered to 50 Å resolution as the initial model in the first 3D classification of each dataset. Initial 3D classification using the closed spike as the initial model was accomplished at bin2 to remove contaminating particles. Particles within 3D classes displaying clear secondary structures were pooled and subjected to cleaning by one round of 2D

classification. For the S/native low-pH dataset, micrographs were imported into cryoSPARC (Punjani et al, 2017) to perform blob picking to obtain a template, then 1,375,445 particles were picked by template picking. One round of 2D classification was followed by the template picking to remove contaminating particles. Selected particles were imported back into RELION to perform an initial 3D classification using a 60 Å filtered closed conformation SARS-CoV-2 S-R/x2 spike map (EMD-11329) as the initial model. Subsequently, a second round of 3D classification using 50 Å filtered closed conformation SARS-CoV-2 S-R/x2 spike map (EMD-11329) was carried out to classify different conformations in each dataset. Auto-refinement, contrast transfer function refinement, and Bayesian polishing were performed iteratively on classified subsets of different conformations.

To classify dynamic features in the Domain D region of locked and closed spikes, we used a previously described focused classification method (Qu et al, 2022) on the S/x1 dataset. Briefly, 576,514 and 944,224 locked and closed spike particles were symmetry expanded in RELION to obtain 1,729,542 and 2,832,672 particles, respectively. A round of in-place 3D classification was carried out using a 30 Å sphere mask around the variable Domain D area using a non-low-pass–filtered consensus map of the region as the reference. Subsequently, each expanded particle was traced back to its original S-trimer particle. The original trimer particles were classified into locked-1, locked-2, locked-112, locked-122, and an asymmetrical closed conformation with extruding FPPR in one protomer. Those classified classes were subjected to another round of 3D auto-refinement.

After the final round of 3D auto-refinement, map resolutions were estimated by the 0.143 criterion using the phase-randomization-corrected Fourier shell correlation curve calculated between two independently refined half-maps multiplied by a soft-edged solvent mask. Final reconstructions were sharpened and locally filtered in RELION. The data-processing procedures were summarized in Fig S4. The estimated B-factors of maps are listed in Table S1.

### Cryo-EM structure model building

SARS-CoV-1 spike ectodomain structure (PDB: 5X58) was used as the starting model for building SARS-CoV-1 spike structures of closed conformations. SARS-CoV-2 spike ectodomain structures (PDB: 7XTZ/7XU2) were used as the starting model for building SARS-CoV-1 structures of locked conformations. PDBs were fitted into maps in UCSF Chimera v1.16 (Pettersen et al, 2004). Manual model rebuilding was performed in Coot 0.9.8 (Casanal et al, 2020). Stereochemistry of the manually built models was optimized using Namdinator (https://namdinator.au.dk) (Kidmose et al, 2019). Final models were refined in PHENIX v1.19.1 (Afonine et al, 2018) by real space refinement with secondary structure restraints and geometry restraints. All structural figures were generated using UCSF Chimera v1.16.

## Data Availability

Cryo-EM density maps for the SARS-CoV-1 S/x1 S-trimer in locked-1, locked-112, locked-122, locked-2, and closed conformations have been deposited in the Electron Microscopy Data Bank (EMDB) with accession codes: EMD-34417, EMD-34418, EMD-34419, EMD-34420,

and EMD-34421; related atomic models have been deposited in the Protein Data Bank (PDB) under accession codes 8H0X, 8H0Y, 8H0Z, 8H10, and 8H11, respectively. Cryo-EM density maps for the SARS-CoV-1 S/x2 S-trimer in closed, locked-2 conformations and SARS-CoV-1 S/x3 S-trimer in locked-1 conformation have been deposited in the EMDB with accession codes: EMD-34422, EMD-34423, and EMD-34424; related atomic models have been deposited in the PDB under accession codes 8H12, 8H13, and 8H14, respectively. Finally, Cryo-EM density maps of SARS-CoV-1 S/native in closed and open conformations determined at low pH have been deposited in the EMDB with accession codes: EMD-34425, EMD-34426; related atomic models have been deposited in the PDB under accession codes 8H15 and 8H16, respectively. Also see Table S1 for data processing and deposition details.

## Supplementary Information

## Acknowledgements

We thank Katarzyna Ciazynska, Andrew Carter (MRC-LMB, Cambridge, UK), and John Briggs (Max Planck Institute of Biochemistry, Martinsried, Germany) for initial design of constructs and development of protein production protocols. We thank Kun Qu (Yong Loo Lin School of Medicine, National University of Singapore) for assistance in cryo-EM data processing. This study was supported by the R&D Program of Guangzhou Laboratory (SRPG22-002 to X Xiong and SRPG22-003 to J He); Emergency Key Program of Guangzhou Laboratory (EKPG21-06 to X Xiong); the Natural Science Fund of Guangdong Province (2021A1515011289 to X Xiong, 2022A1515110495 to J Wang and 2022A1515110211 to B Liu) and the Guangdong-Hong Kong-Macau Joint Laboratory of Respiratory Infectious Diseases (2019B121205010 to J He). X Xiong acknowledges start-up grants from the Chinese Academy of Sciences and Bioland Laboratory (GRMH-GL).

### Author Contributions

X Zhang: investigation, visualization, and writing—original draft, review and editing.
Z Li: investigation, visualization, and writing—original draft, review and editing.
Y Zhang: investigation, visualization, and writing—original draft.
Y Liu: investigation and writing—original draft.
J Wang: funding acquisition and investigation.
B Liu: funding acquisition and investigation.
Q Chen: investigation and methodology.
Q Wang: investigation.
L Fu: investigation and writing—review and editing.
P Wang: resources, supervision, investigation, and writing—review and editing.
X Zhong: investigation.
L Jin: resources, supervision, and investigation.
Q Yan: data curation and investigation.
L Chen: resources, supervision, and investigation.
J He: resources, supervision, funding acquisition, and investigation.
J Zhao: resources, supervision, and investigation.

X Xiong: conceptualization, formal analysis, supervision, funding acquisition, investigation, visualization, project administration, and writing—original draft, review, and editing.

## Conflict of Interest Statement

The authors declare that they have no conflict of interest.

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
