## [Reviewer comments · Life Science Alliance]

Life Science Alliance

Disulfide stabilization reveals conserved dynamic features between SARS-CoV-1 and SARS-CoV-2 spikes.

Xixi Zhang, Zimu Li, Yanjun Zhang, Yutong Liu, Jingjing Wang, Banghui Liu, Qiuluan Chen, Qian Wang, Lutang Fu, Peiyi Wang, Xiaolin Zhong, Liang Jin, Qihong Yan, Ling Chen, Jun He, Jincun Zhao, and Xiaoli Xiong

DOI: <https://doi.org/10.26508/lsa.202201796>

Corresponding author(s): Xiaoli Xiong, Guangzhou Institutes of Biomedicine and Health; Jincun Zhao, Guangzhou Medical University; and Jun He, Guangzhou Institutes of Biomedicine and Health, Chinese Academy of Sciences

Review Timeline:

Submission Date:	2022-11-02
Editorial Decision:	2023-01-23
Revision Received:	2023-05-09
Editorial Decision:	2023-06-08
Revision Received:	2023-06-22
Accepted:	2023-06-22

Scientific Editor: Novella Guidi

Transaction Report:

January 23, 2023

Re: Life Science Alliance manuscript #LSA-2022-01796-T

Xiaoli Xiong
Guangzhou Regenerative Medicine and Health - Guangdong Laboratory, Guangzhou Institutes of Biomedicine and Health,
Chinese Academy of Sciences, Guangzhou, China
Infection and Immunity
190 Kaiyuan Avenue
Guangzhou Science City
Guangzhou, Guangdong 510663

Dear Dr. Xiong,

Thank you for submitting your manuscript entitled "Disulfide stabilization reveals conserved dynamic features between SARS-CoV-1 and SARS-CoV-2 spikes." to Life Science Alliance. The manuscript was assessed by expert reviewers, whose comments are appended to this letter. We invite you to submit a revised manuscript addressing the Reviewer comments.

Thank you for this interesting contribution to Life Science Alliance. We are looking forward to receiving your revised manuscript.

Sincerely,

B. MANUSCRIPT ORGANIZATION AND FORMATTING:

Reviewer #1 (Comments to the Authors (Required)):

In this manuscript, Zhang et al. introduced x1, x2, x3 disulfide bonds into the SARS-CoV spike, which allowed structural characterization of previously unobserved conformations. The x1, x2, x3 disulfides were also utilized in previous studies of the SARS-CoV-2 spike. By cryo-EM SPA method, some newly determined structures of SARS-CoV-1 S were found forming symmetrical locked-1, locked-2 trimers or asymmetrical trimers made up by locked-1 and locked-2 protomers in various combinations. The authors further analyzed and compared very detailed structural features of the SARS-CoV-1 and SARS-CoV-2 spikes in different conformations, including the bound linoleic acid (LA), the domain D that may affect the dynamics different spikes from sarbecoviruses. Several groups have reported the engineered SARS-CoV-2 spike glycoprotein, which could be used as a tool in structural biology, serology, vaccine design and immunology studies. Here the authors extended it to the spike of the SARS-CoV and the results are helpful for understanding the conformational dynamics of the spikes of sarbecovirus including SARS-CoV-1 and SARS-CoV-2.

I think that the results from engineered spikes are especially important for vaccine design and drug development. The authors provided preliminary results about the neutralization and binding by sera from immunized mice. I have two major questions about this part. The first one that the authors did not provide the comparisons between WT and S/x3 immunized sera. Will there be difference between them? This is the most important and interesting point. The second question is about why the authors chose the S/x3 spike for immunization, not S/x1 and S/x2.

The pocket for fatty acid binding in the spikes of sarbecoviruses has been noticed and studied by several different groups. It has also been proposed and explored as a drug-development site for inhibitors locking the spike in the receptor-binding incapable state. The locked conformations of the SARS-CoV-2 seem to prefer the binding of linoleic acid. Why the S/x2 locked-2 does not have the bound linoleic acid? Did the authors have the linoleic acid binding affinity data by SARS-CoV S, SARS-CoV S/x1, SARS-CoV S/x2, SARS-CoV S/x3 and SARS-CoV-2 S?

Other minor issues include:

1. In reducing condition, S/native, S/X1, S/X2 and S/X3 should be the same. Why the behavior of the S/x1 is different from the others in fig1b? It had only one band on the gel.
2. The color of modelled linoleic acid in Fig3 and FigS5 is close to the color of RBD.
3. The authors are suggested to check the text and figures thoroughly. For example, the legend has information about fig. 1d left, middle and right panels. However, the figure 1 has the fig. 1d, fig. 1e and fig. 1f.

Reviewer #2 (Comments to the Authors (Required)):

In the manuscript titled "Disulfide stabilization reveals conserved dynamic features between SARS-CoV-1 and SARS-CoV-2 spikes." Zhang et al. introduced in the SARS-CoV-1 S protein the interdomain disulfides that the group had previously characterized in the SARS-CoV-2 S protein. They determined the structures of these SARS-CoV-2 S protein mutants using cryo-EM and compared the structures with those of the SARS-CoV-2 S proteins and the S proteins of other CoVs. They identified bound ligands in the NTD and RBD regions in their structures. Further, they characterized the immunogenic properties of the engineered spikes.

This is a well-executed study that adds valuable knowledge to our understanding of SARS-CoV-1 S protein conformations and how these relate to the conformations of spikes proteins of other CoV including SARS-CoV-2. The structural studies are well-done, highlighted by the definition of the bound ligands/cofactors.

The nomenclature of the locked forms is a little difficult to understand. It will be useful to the reader if the authors start with an explanation of the nomenclature the first time it is mentioned.

In the "Antigenic and immunogenic properties of engineered spikes" section, it will be useful to see some characterization of the

engineered spikes using known antibodies.

We thank the reviewers for their careful evaluation of the manuscript, we apologise to the reviewers and the editors for the delayed response, this is partly due to a combination of COVID-19 epidemic in China earlier this year, the Chinese New Year and job change of the first author.

Reviewer #1 (Comments to the Authors (Required)):

In this manuscript, Zhang et. al. introduced x1, x2, x3 disulfide bonds into the SARS-CoV spike, which allowed structural characterization of previously unobserved conformations. The x1, x2, x3 disulfides were also utilized in previous studies of the SARS-CoV-2 spike. By cryo-EM SPA method, some newly determined structures of SARS-CoV-1 S were found forming symmetrical locked-1, locked-2 trimers or asymmetrical trimers made up by locked-1 and locked-2 protomers in various combinations. The authors further analyzed and compared very detailed structural features of the SARS-CoV-1 and SARS-CoV-2 spikes in different conformations, including the bound linoleic acid (LA), the domain D that may affect the dynamics different spikes from sarbecoviruses. Several groups have reported the engineered SARS-CoV-2 spike glycoprotein, which could be used as a tool in structural biology, serology, vaccine design and immunology studies. Here the authors extended it to the spike of the SARS-CoV and the results are helpful for understanding the conformational dynamics of the spikes of sarbecovirus including SARS-CoV-1 and SARS-CoV-2.

I think that the results from engineered spikes are especially important for vaccine design and drug development. The authors provided preliminary results about the neutralization and binding by sera from immunized mice. I have two major questions about this part. The first one that the authors did not provide the comparisons between WT and S/x3 immunized sera. Will there be difference between them? This is the most important and interesting point. The second question is about why the authors chose the S/x3 spike for immunization, not S/x1 and S/x2.

We thank the reviewer for his/her comments. Regarding to the first question:

“the authors did not provide the comparisons between WT and

S/x3 immunized sera. Will there be difference between them?”

We thank the reviewer to raise this point, we agree with the reviewer that this point is important and interesting. We have previously studied mouse immune sera raised with different versions of purified SARS-CoV-2 spike proteins (cited as Carnell *et al.* in the current manuscript, PMID: 33963055), including sera raised with S-GSAS-PP (containing a GSAS sequence preventing cleavage at the furin cleavage site, as well as two stabilizing prolines), S-R (changing the furin cleavage site to a single arginine residue to prevent furin cleavage), S-R/PP (changing the furin cleavage site to a single arginine, as well as introducing two stabilizing prolines in S2) and S-R/x2 (changing the furin cleavage site to a single arginine, as well as introducing the x2 disulfide) spike proteins. Among these spike antigens, S-R contains minimum modification to allow expression; S-GSAS-PP is similar to S-R/PP and is a widely used form of antigen in vaccines; S-R/x2 has RBDs restrained in “down” position by the “x2” disulfide and primarily adopts a closed conformation. In Carnell *et al.*, we were not able to find difference of statistical significance among sera in neutralization of pseudoviruses and authentic WT virus. The above antigen raised sera exhibited no difference of statistical significance in binding to the same antigen similar to the SARS-CoV-1 “S/x3” spike raised sera reported in the current manuscript. We did observe, by a Luminex based multiplex antigen binding assay, that among sera raised with RBD and spikes without disulfide stabilization (in these spikes RBDs are not restrained in “down” position), sera show good correlation in antigen binding, meaning that strong binding serum to one antigen also binds strongly to another antigen. However, poor binding correlations were observed when testing the SARS-CoV-2 “S-R/x2” spike raised sera against the other sera, meaning that while certain serum is binding strongly to both “S-R/x2” spike and non-disulfide stabilized spikes or RBD, certain serum is binding strongly only to “S-R/x2” spike but not binding strongly to non-disulfide stabilized spikes or RBD and vice versa. We conclude in Carnell *et al.* that RBD “down” spike sometimes induce rather different antibody response in mice generating sera of distinct binding characteristics (correlations), however, these sera have similar binding and neutralization potencies when compared with sera raised with non-disulfide stabilized spikes. Carnell *et al.* also implicates that, on serology resolution, although we can detect difference in serum binding characteristics it is difficult to understand how the differences come about.

Based on our previous results, we anticipate that SARS-CoV-1 WT “S/native”

and disulfide restrained RBD “down” spikes would behave similarly in mouse immunization as the SARS-CoV-2 versions. Therefore, we do not want to repeat similar works as in Carnell *et al.* In the current context, we simply want to show that a disulfide restrained RBD “down” SARS-CoV-1 spike can induce neutralization sera. We chose “S/x3” because it is more homogenous in conformation compared to “S/x1”. We also chose “S/x3” over “S/x2” because we have demonstrated that SARS-CoV-2 “S-R/x2” induces neutralizing sera. We hope these results should provide useful information to researchers who may be interested in taking investigation further into how spike conformation may affect antibody response.

To make our point clearer, we added “*We have previously shown that “x2” disulfide stabilized, RBD “down”, SARS-CoV-2 S-R/x2 (S-R meaning that the multibasic S1/S2 furin cleavage site is modified to a single arginine) S-trimer induced sera exhibit poor correlation with sera raised by S-trimers with unrestrained RBDs in antigen binding, indicating different antibodies are induced among the RBD “down” S-trimer and S-trimers with unrestrained RBDs, although no difference of statistical significance in neutralization potency among the sera was found (Carnell et al, 2021). It remains poorly understood how a RBD “down” S-trimer induces a different antibody response while generating sera of similar neutralization potencies comparing with S-trimers with unrestrained RBDs. As we have previously demonstrated that “x2” stabilized SARS-CoV-2 S-trimer was able to induce immune sera (Carnell et al., 2021) and “x1” stabilized S-trimer is heterogenous in spike conformation (Fig. 2a-e), purified S/x3 spike was chosen to immunize mice employing a prime-boost immunization strategy.*” in the beginning of the immunization section, we also changed the section title to “*Engineered S/x3 spike induces mouse immune sera*” to avoid overstatement.

The pocket for fatty acid binding in the spikes of sarbecoviruses has been noticed and studied by several different groups. It has also been proposed and explored as a drug-development site for inhibitors locking the spike in the receptor-binding incapable state. The locked conformations of the SARS-CoV-2 seem to prefer the binding of linoleic acid. Why the S/x2 locked-2 does not have the bound linoleic acid? Did the authors have the linoleic acid binding affinity data by SARS-CoV S, SARS-CoV S/x1, SARS-CoV S/x2, SARS-CoV S/x3 and SARS-CoV-2 S?

We thank the reviewer for pointing out current knowledge regarding to fatty acid binding pocket and we have cited the said works in the manuscript as Toelzer et al, 2020 and Toelzer et al, 2022. We first want to address the question “Why the S/x2 locked-2 does not have the bound linoleic acid?” We don’t think we know the exact answer for this question. To elaborate, Zhou et al. (PMID: 33271067) and us (Qu et al., PMID: 33271067) have provided evidences to show that low-pH can promote spike to adopt locked conformation in the absence of lipid bound to RBD; Spike is expressed as a secreted glycoprotein which needs to go through ER and Golgi; ER and Golgi are known to be acidic (PMID: 33157038) and are associated with lipid storage (DOI: 10.5772/intechopen.105450). We have previously speculated in Qu *et al.* (PMID: 33271067) that spikes in acidic ER and Golgi should adopt locked conformations, the locked spike structures determined so far suggest that the lipid binding pocket is blocked by the gating helix in the locked conformations. These observations suggest that lipid binding needs to coordinate with spike folding that lipid needs to be loaded into the lipid binding pocket within RBD before the locked spike trimer is formed. We speculate that the “x2” disulfide may have altered spike folding kinetics and locked “x2” spike is formed before lipid can bind.

“Did the authors have the linoleic acid binding affinity data by SARS-CoV S, SARS-CoV S/x1, SARS-CoV S/x2, SARS-CoV S/x3 and SARS-CoV-2 S?”

We thank the reviewer for this question, unfortunately, we don’t have the linoleic acid binding affinity data. It has been reported by Toelzer et al, 2020 (PMID: 36417532) that RBDs of spikes from diverse betacoronaviruses bind linoleic acid with K_{DS} in the range of 35-87 nM. In the reported assays, the authors used SPR technology and immobilized RBD on the sensor chip. We have considered using similar methods to test lipid binding to S-trimers but we concluded that, in our case, this method is unlikely to be applicable primarily for the following reason: in SPR or BLI methods, the molecular weight ratio of binding ligand:immobilized protein determines the signal strength. The linoleic acid is relatively small compared with immobilized S-trimer in molecular weight therefore lipid binding is unlikely to give strong enough signal to accurately assay binding. The limited amount of protein obtained (around 200 ug from each prep) also prevent us from developing other methods of assaying linoleic acid binding to spike.

Other minor issues include:

-
1. In reducing condition, S/native, S/X1, S/X2 and S/X3 should be the same. Why the behavior of the S/x1 is different from the others in fig1b? It had only one band on the gel.

We thank the reviewer for pointing out this issue. We have performed purification of the S/native, S/x1, S/x2 and S/x3 spikes multiple times, we did notice that the ~120 kDa band varied somewhat in each protein preparations (also see **Fig. S2** in the manuscript). Based on the molecular weight of the band and judged by the fact the band is only obvious when the spike is run as a monomer in the gel we speculate that this band could be the cleavage products (S1 and S2) of SARS-CoV-1 S. SARS-CoV-1 S1 and S2 bands are similar in size (See Figure S5, Walls et. al, PMID: 30712865), when under non-reducing conditions, S1 and S2 are hold together by the stabilizing disulfide bonds.

2. The color of modelled linoleic acid in Fig3 and FigS5 is close to the color of RBD.

We thank the reviewer for his/her suggestion, we changed the colour of modelled linoleic acid to purple, we updated **Fig. 3** and **Fig. S5** accordingly.

3. The authors are suggested to check the text and figures thoroughly. For example, the legend has information about fig. 1d left, middle and right panels. However, the figure 1 has the fig. 1d, fig. 1e and fig. 1f.

We thank the reviewer for his/her careful review, we apologize for the errors. We have revised the **Fig. 1** legend to rectify the errors.

Reviewer #2 (Comments to the Authors (Required)):

In the manuscript titled "Disulfide stabilization reveals conserved dynamic features between SARS-CoV-1 and SARS-CoV-2 spikes." Zhang et al. introduced in the SARS-CoV-1 S protein the interdomain disulfides that the group had previously characterized in the SARS-CoV-2 S protein. They determined the structures of these SARS-CoV-2 S protein mutants using cryo-EM and compared the structures with those of the SARS-CoV-2 S proteins and the S proteins of other CoVs. They identified bound ligands in the NTD and RBD regions in their structures. Further, they characterized the immunogenic properties of the engineered spikes.

This is a well-executed study that adds valuable knowledge to our understanding of SARS-CoV-1 S protein conformations and how these relate to the conformations of spikes proteins of other CoV including SARS-CoV-2. The structural studies are well-done, highlighted by the definition of the bound ligands/cofactors.

The nomenclature of the locked forms is a little difficult to understand. It will be useful to the reader if the authors start with an explanation of the nomenclature the first time it is mentioned.

We thank the reviewer for his/her comments and we apologize for difficulty caused in understanding the nomenclature regarding the locked forms.

We have reworded the sentences when locked conformation is first introduced in introduction section of the manuscript:

“However, a third prefusion conformation, designated “locked”, was subsequently identified in multiple studies using either full-length spikes or spike ectodomains (Bangaru et al, 2020; Cai et al, 2020; Toelzer et al, 2020; Wrobel et al, 2020; Xiong et al, 2020). S-trimer in locked conformation features 3 “down” RBDs. Different from S-trimer in closed conformation which also have 3 “down” RBDs, locked S-trimer is structurally more ordered, adopting a more tightly packed trimeric quaternary structure, and usually possessing features including, bound lipid in RBD, rigidified Domain D region and ordered fusion peptide proximal region (FPPR, residues 833-855 in SARS-CoV-2 S) (Qu et al, 2022; Xiong et al., 2020). Interestingly, locked conformation appears to be transient, in pH neutral phosphate buffered saline (PBS), only a small fraction of purified spikes adopted locked conformations (Xiong et al., 2020).”

We added more info about “locked-1” and “locked-2” conformations in the introduction:

“Structural studies of x3 disulfide stabilized locked spikes allowed us to further classify locked spikes into the originally identified “locked-1” (Bangaru et al., 2020; Cai et al., 2020; Toelzer et al., 2020; Wrobel et al., 2020; Xiong et al., 2020) and additionally identified “locked-2” (Qu et al., 2022) conformations. The two conformations differ primarily in ways how Domain D region rigidifies. We observed that in the “locked-1” conformation Domain D is rigidified with a large, disordered Domain D-loop, while in the “locked-2” conformation Domain D is rigidified with the Domain D-loop fully ordered.

Refolding of Domain D-loop is observed between the two locked conformations resulting some Domain D residues engaging in different interactions (Qu et al., 2022)."

We hope the above rewritings should improve understanding of the nomenclature regarding the locked forms.

In the "Antigenic and immunogenic properties of engineered spikes" section, it will be useful to see some characterization of the engineered spikes using known antibodies.

We thank the reviewer for this suggestion, we have curated 19 structurally characterized mAbs known to bind SARS-CoV-1 S from the CoV-AbDb (<https://opig.stats.ox.ac.uk/webapps/covabdab/>), we downloaded their PDBs and structurally aligned them to the S/x3 structure which has 3 RBDs in "down" position, we found that 17 out of the 19 mAbs would clash when binding to their epitopes within the RBD "down" SARS-CoV-1 S-trimer. Only 2 mAbs, namely CV38-142 and S309, are able to bind RBD "down" SARS-CoV-1 S-trimer without clash. We and others have previously shown that spike conformation can affect binding of mAbs targeting cryptic or semi-cryptic epitopes (meaning epitopes completely or partly obstructed in RBD "down" spike respectively) (Qu et al., PMID: 33271067; He et al., PMID: 36151403; McCallum et al., PMID: 32753755).

We included a new **Fig. S11** to illustrate binding of mAbs to RBD "down" S-trimer. In the **Fig. S11** legend, we included the results of the above analysis:

"Fig. S11 | Binding of many known SARS-CoV-1 S binding antibodies to RBD "down" x3 SARS-CoV-1 S-trimer results in steric clash. 19 structurally characterized mAbs reported to bind SARS-CoV-1 were curated from CoV-AbDb (Raybould et al, 2021), namely CV38-142 (Liu et al, 2021), S309 (Pinto et al, 2020), F26G19 (Liu et al., 2021), DH1047 (Martinez et al, 2022), ADI55688 (Yuan et al, 2022), m396 (Prabakaran et al, 2006), 80R (Hwang et al, 2006), 47D11 (Fedry et al, 2021), 553-15 (Zhan et al, 2022), C118 (Jette et al, 2021), CR3022 (Wu et al, 2020), COVA1-16 (Liu et al, 2020), GW01 (Wang et al, 2022), S304 (Piccoli et al, 2020), H014 (Lv et al, 2020), S230 (Walls et al, 2019), VHH-72 (Wrapp et al, 2020a), Nb70 (Li et al, 2022), DH1058 (Gobeil et al, 2022). In each panel, Fab is coloured in pink, RBD is coloured in blue and fusion peptide (FP) is coloured in orange. Structures of mAbs in complex with their respective antigens are structurally aligned onto the currently reported 3-RBD "down" S/x3 locked-1 structure (PDB: 8H14). This analysis identifies that only CV38-142 (PDB: 7LM9 (Liu et al., 2021)) and S309 (PDB: 8DW2

(Chonira et al, 2023)) are able to bind RBD “down” spike without clash. Other mAbs are not able to bind RBD “down” spike without clash. Previous studies have shown that binding of mAbs to epitopes completely or partly obstructed in RBD “down” spike is strongly affected by spike conformation (He et al, 2022; McCallum et al., 2020; Qu et al., 2022).”

We referred the new **Fig. S11** in the main text in the discussion section with the following sentence:

*“A survey of structurally characterized SARS-CoV-1 S binding monoclonal antibodies (mAbs) identify that only 2 out 19 of the structurally characterized SARS-CoV-1 S binding mAbs are able to bind RBD “down” S-trimer without clash (**Fig. S11**).”*

To avoid overstatement, we changed the section title of "Antigenic and immunogenic properties of engineered spikes" section to “Engineered S/x3 spike induces mouse immune sera”

Other Modifications:

We updated reference “Toelzer et al, 2022” from preprint version to the published version.

We included a new author “Qihong Yan” for his role in curating SARS-CoV-1 S binding mAbs.

We updated the Acknowledgement and author contributions sections.

June 8, 2023

RE: Life Science Alliance Manuscript #LSA-2022-01796-TR

Dr. Xiaoli Xiong
Guangzhou Institutes of Biomedicine and Health
Infection and Immunity
190 Kaiyuan Avenue
Guangzhou Science City
Guangzhou, Guangdong 510663
China

Dear Dr. Xiong,

Thank you for submitting your revised manuscript entitled "Disulfide stabilization reveals conserved dynamic features between SARS-CoV-1 and SARS-CoV-2 spikes.". We would be happy to publish your paper in Life Science Alliance pending final revisions necessary to meet our formatting guidelines.

- please upload your manuscript text as a doc file
- please upload both your main and supplementary figures as single files and add a separate figure legend section to your main manuscript
- please make sure your table files are uploaded as excel or doc files
- please add ORCID ID's for corresponding authors-you should have received instructions on how to do so
- please add a summary blurb and a category for your manuscript to our system
- please add the Twitter handle of your host institute/organization as well as your own or/and one of the authors in our system
- please add a conflict of interest statement to your main manuscript text
- please consult our manuscript preparation guidelines <https://www.life-science-alliance.org/manuscript-prep> and make sure your manuscript sections are in the correct order
- please remove the separate Importance section
- please add a figure callout for Figure S9a-f to your main manuscript text
- please include the appropriate approval information for the mouse work, and who granted that approval

A. FINAL FILES:

B. MANUSCRIPT ORGANIZATION AND FORMATTING:

Sincerely,

Reviewer #1 (Comments to the Authors (Required)):

In the rebuttal letter, the authors have carefully answered the questions raised by the reviewers. For some questions, they also modified the manuscript based on the answers to the reviewers' questions and suggestions. Overall, I am satisfied with the revisions and think that the revised manuscript could be accepted and published directly.

Reviewer #2 (Comments to the Authors (Required)):

The authors have addressed all my critiques satisfactorily.

June 22, 2023

RE: Life Science Alliance Manuscript #LSA-2022-01796-TRR

Dr. Xiaoli Xiong
Guangzhou Institutes of Biomedicine and Health
Infection and Immunity
190 Kaiyuan Avenue
Guangzhou Science City
Guangzhou, Guangdong 510663
China

Dear Dr. Xiong,

Thank you for submitting your Research Article entitled "Disulfide stabilization reveals conserved dynamic features between SARS-CoV-1 and SARS-CoV-2 spikes.". It is a pleasure to let you know that your manuscript is now accepted for publication in Life Science Alliance. Congratulations on this interesting work.

DISTRIBUTION OF MATERIALS:

Again, congratulations on a very nice paper. I hope you found the review process to be constructive and are pleased with how the manuscript was handled editorially. We look forward to future exciting submissions from your lab.

Sincerely,
